# GestroNet: A Framework of Saliency Estimation and Optimal Deep Learning Features Based Gastrointestinal Diseases Detection and Classification

**DOI:** 10.3390/diagnostics12112718

**Published:** 2022-11-07

**Authors:** Muhammad Attique Khan, Naveera Sahar, Wazir Zada Khan, Majed Alhaisoni, Usman Tariq, Muhammad H. Zayyan, Ye Jin Kim, Byoungchol Chang

**Affiliations:** 1Department of Computer Science, HITEC University, Taxila 47080, Pakistan; 2Department of Computer Science, University of Wah, Wah Cantt, Rawalpindi 47040, Pakistan; 3Computer Sciences Department, College of Computer and Information Sciences, Princess Nourah bint Abdulrahman University, Riyadh 11671, Saudi Arabia; 4Department of Management Information Systems, CoBA, Prince Sattam Bin Abdulaziz University, Al-Khraj 16278, Saudi Arabia; 5Computer Science Department, Faculty of Computers and Information Sciences, Mansoura University, Mansoura 35516, Egypt; 6Department of Computer Science, Hanyang University, Seoul 04763, Korea; 7Center for Computational Social Science, Hanyang University, Seoul 04763, Korea

**Keywords:** stomach diseases, contrast enhancement, saliency estimation, Bayesian optimization, features optimization

## Abstract

In the last few years, artificial intelligence has shown a lot of promise in the medical domain for the diagnosis and classification of human infections. Several computerized techniques based on artificial intelligence (AI) have been introduced in the literature for gastrointestinal (GIT) diseases such as ulcer, bleeding, polyp, and a few others. Manual diagnosis of these infections is time consuming, expensive, and always requires an expert. As a result, computerized methods that can assist doctors as a second opinion in clinics are widely required. The key challenges of a computerized technique are accurate infected region segmentation because each infected region has a change of shape and location. Moreover, the inaccurate segmentation affects the accurate feature extraction that later impacts the classification accuracy. In this paper, we proposed an automated framework for GIT disease segmentation and classification based on deep saliency maps and Bayesian optimal deep learning feature selection. The proposed framework is made up of a few key steps, from preprocessing to classification. Original images are improved in the preprocessing step by employing a proposed contrast enhancement technique. In the following step, we proposed a deep saliency map for segmenting infected regions. The segmented regions are then used to train a pre-trained fine-tuned model called MobileNet-V2 using transfer learning. The fine-tuned model’s hyperparameters were initialized using Bayesian optimization (BO). The average pooling layer is then used to extract features. However, several redundant features are discovered during the analysis phase and must be removed. As a result, we proposed a hybrid whale optimization algorithm for selecting the best features. Finally, the selected features are classified using an extreme learning machine classifier. The experiment was carried out on three datasets: Kvasir 1, Kvasir 2, and CUI Wah. The proposed framework achieved accuracy of 98.20, 98.02, and 99.61% on these three datasets, respectively. When compared to other methods, the proposed framework shows an improvement in accuracy.

## 1. Introduction

Cancer is a deadly disease that is currently the leading cause of death worldwide [1]. Cancer is caused primarily by genetics, but it is also influenced by environmental factors. Environmental factors such as a person’s eating habits and community behaviors are the leading causes of 50% of cancer cases. This disease typically develops after 20–30 years of exposure to harmful cancer-causing agents [2]. Modern medical technologies are being used to detect cancer in its early stages, with radical resection being used to treat approximately 50% of cases. The stomach is a vital muscular part of the body that aids in food digestion. It is positioned to the left of the upper abdomen. The food is removed from the esophagus via a muscular valve known as the gastroesophageal sphincter [3]. The stomach is responsible for three major functions. The first and most important task is to store the food that we consume. Furthermore, by releasing gastric juices, it breaks down, and digests the food consumed. Finally, the digested food is transferred to the small intestine. Stomach infection, also known as gastric cancer, is one of the fourth leading causes of cancer deaths worldwide, with an average survival ratio of less than 12 months for advanced cancer stages. It is a polygenic disease in which multiple factors, both genetic and environmental, play an important role in its development. Every year, an estimated 1 million new cases are identified worldwide [4]. Gastric cancer cannot be avoided; if the warning signs are ignored or not treated in the early stages, it can develop into a tumor. Most treatments for it include chemotherapy, surgery, and radiation, among other things. Men have a higher risk of developing gastric cancer compared to women. Aside from environmental and genetic factors, Helicobacter pylori (H. pylori) are a type of bacteria that enters the body and lives in the digestive region, resulting in ulcers and gastritis over time. Gastritis is an inflammation of the stomach lining caused by the same bacterium that causes stomach ulcers. Gastritis symptoms include hiccups, heartburn, and gagging [5]. 

The most common gastric disease is colorectal cancer, which is the third most common cancer and affects both men and women equally [6]. Colorectal cancer, also known as bowl cancer, is characterized by three infections: bleeding, ulcer, and polyp. According to statistics, approximately 1.6 million people suffer from aching bowl infections, and approximately 200,000 new cases of colorectal cancer are diagnosed each year [7]. These gastrointestinal infections (GIT) can be cured if detected and treated early. The diagnosis of GIT, particularly at an early stage with improved accuracy, has emerged as the focus of current research [8]. 

Endoscopy is an effective method for identifying gastric cancer and is one of several diagnostic measures that could be used to detect it. The accuracy of endoscopy biopsy removal is approximately 98%; however, this procedure is time consuming, costly, requires trained medical specialists, and if not performed correctly, can result in multiple complications in the patient [9]. Another treatment for gastric cancer diagnosis is gastroscopy and laparoscopy, in which a camera is inserted into a patient’s esophagus and stomach and an analysis is made using double contrast imaging of the stomach. The majority of gastrointestinal infections can lead to colorectal cancer, which manifests as short bowl syndrome and hemorrhoids. Wireless capsule endoscopy (WEC) is a painless method of identifying infections such as ulcers and polyps in the gastrointestinal tract (GIT) areas of patients with limited access without the need for surgery [10]. In this procedure, the patient is asked to swallow a camera in the shape of a capsule with dimensions of 11 × 30 mm^2^ and no exterior wiring, which captures images and sends them to a data recorder via an RF transmitter as it travels to the gastrointestinal tract (GIT). The video frames captured have a resolution of 255 × 255 pixels and are compressed in jpeg format. This entire process takes an average of 120 min to complete, whereas in normal circumstances it takes approximately 2 h [11].

An average of 60,000 images of a single patient are manually analyzed, which is sometimes impossible for even a qualified doctor to carry out [12]. Though analysis of all image frames is not required, in order to obtain accurate results, the doctor evaluates all captured frames, which lead to a major disagreement. To address this, specialists have used a variety of computer aided diagnosis (CAD) techniques to help identify gastrointestinal tract (GIT) infections using wireless capsule endoscopy (WCE) images. However, selecting image frames containing abnormalities is a difficult task due to the similarity of signs containing texture, shape, and color [13], which makes accurately categorizing the nature of infection difficult. To bridge this gap in computer aided diagnosis (CAD) systems, various researchers used computational pathology techniques and algorithms from image processing (IP), artificial intelligence (AI), machine learning (ML), and deep learning (DL), with promising results [14]. Deep learning applications have yielded promising results in the classification of cancer, cell segmentation, and predicting the outcome of several gastrointestinal infections in recent years (GI) [15].

A significant amount of work has been done in the field of medical imaging to assist general practitioners in the accurate identification and classification of human diseases such as breast tumor, lung cancer, brain tumor [16], and infections linked to the stomach such as gastrointestinal tract (GIT) using wireless capsule endoscopy (WCE) images [17]. The stomach is an important organ in the human body. Gastritis, ulcers, polyps, and bleeding are all examples of harmful stomach diseases. Khan et al. [18] presented a deep learning architecture for the detection and classification of gastrointestinal tract (GIT) anomalies. The entire procedure is evaluated using two datasets: Private and KVASIR. When compared to the existing techniques, the proposed technique proves to be more effective. The accuracy achieved for the private dataset is 99.8%, while the accuracy for the KVASIR dataset is 86.4%. However, the study also addresses the disadvantage of varying texture features for disease classification. A feature optimized DL technique for the classification of gastrointestinal infections using wireless capsule endoscopy (WCE) images is proposed in the study [19]. The proposed technique is evaluated using two publicly available datasets. The assessed results’ accuracy is 99.5%, which is more effective when compared to existing methods and distinct optimal feature sets. However, the study also mentions the applied fusion method’s shortcoming of being time consuming, which can be overcome in future studies by building a CNN model from the start. Khan et al. [20] presented a computerized automated system for classifying gastrointestinal tract (GIT) infections from wireless capsule endoscopy (WCE) images using a robust deep CNN feature selection method. Infected areas are segmented using the CFbLHS method before CNN features are computed. Furthermore, only the best features were chosen for the final classification. Private datasets are used to conduct the experiment, resulting in a maximum accuracy of 99.5% over a computational time period of 21.15 s. The shortcomings of manual procedures for identifying gastric infections are overcome by the use of various high-tech practices that help physicians detect gastric abnormalities using WCE images. The study [21] envisions a fully computerized deep learning feature fusion centered architecture for the classification of numerous gastric anomalies. For evaluating the results, a database for wireless capsule endoscopy (WCE) images is created, resulting in a maximum accuracy of 99.46% when compared to the existing techniques. The overall results show that the preprocessing phase of the CNN model is effective in the learning procedure, and that the fusion of optimal features improves the accuracy. However, unrelated features and redundancy were still observed. A case study was conducted to assess the pathologist’s deep learning competence in diagnosing gastric infections [22]. In this study, 16 professional pathologists inferred a total of 110 whole slide images (WSI) containing 50 malignant and 60 benign tumors with and without deep learning assistance. This case study concluded that deep learning-based assistance aided in achieving maximum area curvature under ROC-AUC, higher sensitivity, and a normal analysis time span when compared to unassisted deep learning. As a result, it was determined that deep learning-based assistance was effective, accurate, and efficient in the detection of gastric tumors. Majid et al. [23] presented an automated technique for identifying and classifying stomach infections using endoscopic images. The technique is divided into four phases: feature extraction, feature fusion, feature selection, and classification. The proposed method is evaluated using a database comprised of four datasets: CVC-ClinicDB, KVASIR, and ETIS-LaribPolypDB. The evaluation results show that the features selection method performs well by refining the overall computational time period, with a maximum accuracy of 96.5%. Authors in [24] presented a technique for assessing and classifying gastric ailments using wireless capsule endoscopy (WCE) images that cover the four stages. The first phase employs HSI color modification, which is followed by the infection segmentation phase via the saliency approach. For image fusion in the third phase, a probability method is used. Finally, traditional features are extracted and classified by machine learning methods. Al-Adhaileh et al. [25] presented a method for the detection of gastrointestinal contagions. Three deep learning based models have been employed such as AlexNet, GoogleNet, and ResNet-50 that can aid medical practitioners to concentrate on the areas which have been overlooked during diagnosis. KVASIR dataset is used for the evaluation process and achieved an accuracy of 97%. 

Preprocessing of original images, segmentation of cancer region, feature extraction, and classification are all important steps in a CAD system. Preprocessing is an important step in medical image processing because it allows important information to be visualized more effectively. This step improves the segmentation process, which has a later impact on the extraction of accurate features. However, segmentation of the infected stomach region is difficult due to the change in shape and the presence of the infected region on the boundary region. Incorrect segmentation of infected regions not only reduces segmentation accuracy but also raises the error rate in the classification phase due to the extraction of irrelevant features. Feature extraction is an important step in accurate classification, but several redundant features are sometimes extracted. The main reason is that deep models are trained using raw images or inaccurately segmented images. Furthermore, recent studies show that deep models were trained on static hyperparameters, which can be improved by using a dynamic initialization approach. To address the issue of redundant features, researchers developed several feature selection methods, including genetic algorithm-based selection, ant colony-based selection, entropy-based selection, and others. The best features are selected using these techniques; however, based on the analysis of these methods, it is also discovered that some important features are also removed during the selection process. In this paper, we proposed deep saliency estimation and an optimal deep features-based framework for classifying stomach infections. The following are our major contributions: We proposed a hybrid sequential fusion approach for contrast enhancement. The purpose of this approach is improving the contrast of infected region in the image that further helps in better segmentation.A deep saliency-based infected region segmentation and localization technique is proposed.A fine-tuned MobileNet-V2 model is trained on localized images and hyperparameters are optimized using Bayesian Optimization. Usually, the hyperparameters were initialized in a static way.A hybrid whale optimization algorithm is proposed for the selection of best features.

## 2. Materials and Methods

A new automated framework is proposed in this work for GIT diseases detection and classification using saliency estimation and Bayesian optimization deep learning features. The proposed framework consists of several steps that include preprocessing to classification. Original images are improved in the preprocessing step that further segmented through saliency map estimation and mathematical formulation. The segmented regions are then used to train a pre-trained fine-tuned model called MobileNet-V2 using transfer learning. The fine-tuned model’s hyperparameters were initialized using Bayesian optimization (BO) and extract features from average pooling layer. After that, we proposed a hybrid whale optimization algorithm for the selection of best features. Finally, the selected features are classified using an extreme learning machine classifier. Figure 1 shows the overall framework of GIT diseases segmentation and classification. The detail of each step is provided in the below subsections.

### 2.1. Proposed Contrast Enhancement

Contrast enhancement is an important step in the domain of image processing based on some important properties such as low contrast improvement and noise estimation. In this work, we proposed a hybrid approach for contrast enhancement that based on some fusion of sequential steps. Consider, we have a database denoted by Δ having several images of different classes. Let ϕ(i,j) is an input image of dimension 256×256×3 and $\tilde{\phi }$ is a resultant image. For this, initially a CNN-based denoising network is employed and the bubble regions in the image is removed. This process is defined as follows: (1)$$I_{1}=\frac{c_{i}}{\sum_{i=1}^{3}\left(\tilde{C}\right)},\;\tilde{C}\in \left(c_{1},c_{2},c_{3}\right)$$
(2)$$f_{1}=\varphi \left(c_{1},\;\tilde{\varphi }\right)$$
(3)$$f_{2}=\varphi \left(c_{2},\;\tilde{\varphi }\right)$$
(4)$$f_{3}=\varphi \left(c_{3},\;\tilde{\varphi }\right)$$
(5)$$f_{4}=cat\left(3,\;f_{1},f_{2},f_{3}\right)$$
where $\tilde{\varphi }$ is pre-trained network [26], $f_{4}$ is a denoise image, and $c_{i}$ denotes the extracted three channels, respectively. In the next step, top-bottom hat filtering is applied on $f_{4}$ to improve the local and global information as follows:(6)$$\phi_{top}=Top\left(f_{4},s\right)$$
(7)$$\phi_{bot}=Bot\left(f_{4},s\right)$$
(8)$$\phi_{fused}=\left(\phi_{top}+\phi_{bot}\right)-\phi \left(i,j\right)$$

This resultant image has some brightness effects that are resolved through haze removal existing approach [26]. The output can be written as follows: (9)$$\phi_{hz}=HZ\left(\phi_{fused}\right)$$

For highlighting the important information in the image, we performed multiplication operation that was finally fused with $\phi_{hz}$ image for final output.
(10)$$\phi_{ml}=\phi_{hz}*\phi_{nw}$$
(11)$$\phi_{nw}=\phi_{fused}*\phi_{top}$$
(12)$$\tilde{\phi }=\sum \left(\phi_{ml}+\phi_{hz}\right)$$

The resultant $\tilde{\phi }$ is visually shown in Figure 2. 

### 2.2. Proposed Saliency Map Based Segmentation

Saliency-based segmentation of an object is a new domain of research in the imaging and medical domain. In medical domain, the saliency-based segmentation of infected region is a new challenge and can perform better than the traditional techniques such as thresholding, clustering etc. In this work, we proposed a deep saliency based segmentation method called deep saliency map of infected region. The proposed technique works in four serial steps. In the first step, we design a 14 layered CNN architecture and trained on enhanced images. In the second step, weights of second convolutional layer were visualized and merged into a single image. In the third step, thresholding-based convert image into a binary form that refined in the last step by employing some mathematical operations such as closing and filling. 

The newly designed 14 layered CNN architecture includes three convolutional layers having filter size 3×3 and stride 2×2, 2 max pooling layers having filter size 2×2 and stride 2×2, 3 batch normalization layers, 3 activation layers (ReLu), one average pooling layer, one fully connected layer, and last one is Softmax layer (can be seen in Figure 3). This designed architecture is trained on enhanced images, whereas the learning rate is 0.05, epochs are 100, mini batch size is 32, momentum value is 0.6, dropout factor is 0.5, and stochastic gradient descent (SGD) is employed as optimizer. After the training, we visualized the weights of second convolutional layer and merge all those sub-images which have clear pattern into a single image. 

The resultant image is further refined by Equation (13) to improve the visual map of infected region as follows: (13)$${\tilde{O}}_{map}=\left[O_{map}+\sum_{i=1}c_{1}\left(\tilde{\phi }\right)\right]$$

Here, ${\tilde{O}}_{map}$ denotes the refined infection saliency map image, as illustrated in Figure 4, $O_{map}$ denotes the original saliency mapped image, $\tilde{\phi }$ is proposed enhanced image. After that, the resultant refined saliency mapped image is converted into a binary form by employing the following equation: (14)$$\phi_{binary}=\{\begin{array}{c} 1\;\;\;\;\;\;\;\;if\;\;{\tilde{O}}_{map}\geq t\;\\ 0\;\;\;\;\;\;\;\;\;\;\;\;Otherwise \end{array},\;\mathrm{where}\;t=Avg\;Value$$

The proposed binary image $\phi_{binary}$ as shown in Figure 4 is further refined by mathematical operations such as filling and closing. The effects after the closing and filling operations are shown in Figure 4. The final refined binary image is further localized through active contour approach and infected regions are later utilized for the training of a deep model. A few sample localized images have been illustrated in Figure 5. 

### 2.3. Deep Learning Features

In this work, we extracted deep learning features for the infected region classification. For deep features extraction, a pre-trained MobileNet-V2 [27] deep model is employed. This model contains around 154 layers including convolutional layers, pooling layers, and fully connected. This model is specifically designed for the classification and general feature generation of images. This model uses 3.4 M constraints that are fewer than other generally preferred models of convolutional neural network (CNN) for example AlexNet uses 61 M, ResNet50 uses 23 M constraints, VGGNets uses 138 M, and GoogleNet uses 7 M constraints. Originally, the output layer of this model consists of 1000 object classes. We fine-tuned this model and replaced the last layers with new layers according to the output classes of proposed framework such as classes in selected datasets. The training is performed through deep transfer learning, as shown in Figure 6. In this figure, it is noted that the knowledge is transferred to the fine-tuned model. In the training process, usually static hyperparameters have been initialized but in this work, we employed Bayesian optimization for the initialization of hyperparameters. The following hyperparameters are initialized through BO- learning rate (0.0001–1), momentum (0.6–0.8), and L2-Regularization ($1e^{-10},\;1e^{-2}$). After that, a newly trained model is obtained that includes GIT disease classes. 

### 2.4. Bayesian Optimization

For the purpose to optimize the expensive noisy tasks of black box, Bayesian optimization [28] is employed in this work. A substantial improvement with notional outcomes has been stated by the current revolution in Bayesian optimization. The strategy of BO relies over the assembly of heuristic model on which several objective tasks are disseminated to the objective of concern from the input space.
(15)$$D=\left\{\left(a_{x,\;}b_{x}\right)\right\}\;\begin{array}{c} N\\ x=1 \end{array}$$
where N is the over-all sum of annotations of the input objective sets. An acquisition function is applied to the variance and mean which is an interchange between exploitation and exploration. In order to decide the next input for assessment, a proxy optimization is executed by continuing the Bayesian optimization. Functions used in BO are disseminated using GPs due to flexibility, ambiguity, and systematic properties. Hence to overcome minimization complications, BO is utilized and defined as follows:(16)$$y^{*}=\begin{array}{c} \mathrm{argmin}\\ y\in \;X \end{array}g(y)$$

In the above stated equation, X denotes the dense subset of ℝK. For the meta-parameters of substitute model, let borderline analytical variance of heuristic model be σ2(y,Θ) = ∑(y, y; Θ) and μ(y; D, Θ) that represents the analytical mean and it is described as follows: (17)γ(y)=g(yBEST)−μ(y; D, Θ) σ(y; D, Θ) 
where g(yBEST) represents the minimum perceived value. The estimated enhanced benchmark is shown as:(18)αFI(y; D, Θ) =σ(y; D, Θ)·[γ(y)Φ (γ(y))+M (γ(y);0,1)]

Here, the symbol Φ represents the cumulative function and M(.;0, 1) signifies the density of normal standard. After the training of this model on infected cropped regions, we obtained a newly trained model that is later utilized for the features extraction. The features are extracted from the average pooling layer of dimension N×1280. As illustrated in Figure 1, the extracted features are optimized through a hybrid whale optimization algorithm.

### 2.5. Proposed Feature Selection Algorithm

Feature selection is a vital preprocessing phase in machine learning. The collection of huge quantity of data and information may result into noise and irrelevant data which in turn impacts the accuracy of the system. Feature selection is a significant approach that selects the best features from the original feature matrix. The purpose of this step is to ignore the redundant information and minimize the computational time. In this work, we proposed a hybrid optimization algorithm based on Whale optimization [29] and Harris Hawks optimization [30]. The mean deviation formulation is applied after both algorithms to remove redundancy among features. Mathematically, the Whale optimization algorithm is defined as follows: (19)$$\vec{X}=\vec{U}.\vec{Y}\;\left(v\right)-\vec{Y}\left(v\right)$$
(20)$$\vec{Y}\left(v+1\right)=\vec{\;Y}(v)-\vec{W}.X$$

In the above stated equations, $\vec{\;Y}$^*^ implies the top most attained result, whereas v signifies the number of repetitions. The symbols $\vec{W}$ and $\vec{U}$ are constants and defined as follows:(21)$$\vec{W}=2\;\vec{c}.\;\vec{d}-\vec{c}$$
(22)$$\vec{U}=\;2.\vec{d}$$
where $\vec{d}$ implies random vector among [0, 1], $\vec{c}$ is also a random vector which is meant to regulate the overall conjunction method and it declines linearly from 2 to 0 during the repetitions. The value of $\vec{c}$ can be calculated using the below equation.
(23)$$\vec{c}=2-v\frac{2}{V}$$
where v implies the current repetition and V implies the large number of repetitions. The transition course between exploration and exploitation is shown by vector $\vec{W}$. The exploration agents will keep on exploring the space when the absolute value of vector $\vec{W}$ is greater than one. Moreover, when absolute value of vector $\vec{W}$ is less than one i.e., $|\vec{W}|<1$, it will result in exploitation of the solution. The two major approaches included in this algorithm are known as encircling technique and spiral shaped technique. The encircling process can be attained by minimizing the value of W, whereas the second process update the distance between the current search agent which is attained at point $\vec{\;Y}$^*^ and the exploration agent. Mathematically, it is formulated as follows:(24)$$Y\;\left(v+1\right)=X.eal.\;cos\left(2\pi l\right)+Y^{*}\left(v\right)\;\;\;$$

A probability value of 0.5 is set for the purpose of signifying the explorative behavior for further execution. This procedure is stated below as:(25)$$\vec{Y}\left(v+1\right)=\{\begin{array}{c} X^{l}.e^{al}.\cos\left(2\pi l\right)+Y^{*}\left(v\right),\;\;P\geq 0.5\\ \;{\vec{Y}}^{*}\left(v\right)-\vec{W}.X\;,\;\;\;\;\;\;\;\;\;\;\;\;\;\;\;\;\;\;\;P\leq 0.5 \end{array}$$

In Equation (25), the probability value of 0.5 is selected for the selection of final features but after the analysis of this formulation on different probability values, we observed that the static value is not a good choice; therefore, we modified this equation by employing a median value of selected features instead of 0.5. Hence, the above equation can be written as: (26)$$\vec{Y}\left(v+1\right)=\{\begin{array}{c} X^{l}.e^{al}.\cos\left(2\pi l\right)+Y^{*}\left(v\right),\;\;P\geq MD\\ \;{\vec{Y}}^{*}\left(v\right)-\vec{W}.X\;,\;\;\;\;\;\;\;\;\;\;\;\;\;\;\;\;\;\;\;P\leq MD \end{array}$$
(27)$$MD=\{\begin{array}{c} X^{*}\left[\frac{n+1}{2}\right]\;\;\;\;\;\;\;if\;\;\;\;\;\;\;\;\;\;\;\;n\;is\;odd\\ \frac{X^{*}\left[\frac{n}{2}\right]+X^{*}\left[\frac{n}{2}+1\right]}{2}\;\;\;if\;n\;is\;even \end{array}$$
where MD denotes median value and $X^{*}$ denotes the current iteration features utilized for the final selection through Equation (25). Through Equation (25), the dimension of selected features is N×728. These features are passed to Harris Hawks optimization for one extra step refinement. Harris Hawks approach is replicated by the system in two states. During first state, this approach settles at several random localities nearby their family or cluster. In the second state, Harris Hawks can live around other supporters of household or cluster, Q ≥ MD is for the first scenario and for later state Q < MD.
(28)$$Y\;\left(v+1\right)=\{\begin{array}{c} Y_{RAND}-d_{1}\left|Y_{rand}\left(v\right)-2d_{2}Y\left(v\right)\right|,\;\;\;\;\;\;\;\;\;\;\;\;\;\;\;\;\;\;\;\;\;\;\;\;\;\;\;Q\;\geq \;Mean\\ (Y_{RABBIT}\left(v\right)-Y_{M}\left(v\right)-d_{3}\left(lb+d_{4}\left(ub-lb\right)\right),\;\;\;Q<Mean \end{array}$$

In the above stated question, Y (v+1) represents the location of succeeding hawk where as $Y_{rand}\left(v\right)$ represents the current point of the selected hawk from prevailing population. The mean value-based selection is performed instead of 0.5 static values. More information can be found here [30]. The ELM classifier is selected as a fitness function and fitness value is computed based on the error rate. This process was continued for initialized iterations such as 200 in this work. At the end, we obtained a final selected feature vector of dimension N×624 that was finally classified using extreme learning machine (ELM) classifier.

## 3. Results

The proposed framework experimental process is conducted in this section in the form of numerical values and plots. Three publicly available datasets have been employed for the experimental process such as Kvasir 1 [31], Kvasir 2 [32], and CUI Wah [33]. The Kvasir datasets consists of eight different classes as illustrated in Figure 7. In the Kvasir V1, each class includes 500 images (total 4000 images), whereas the Kvasir V2 dataset includes 1000 images in each class (total 8000 images). The 50:50 approach was opted for training and testing of deep models. The cross validation opted for value 10. Several classifiers have been utilized for the comparison of ELM classifier accuracy such as fine tree, quadratic SVM (Q-SVM), weighted KNN (W-KNN), and bi-layered neural network (Bi-Layer NN). The performance of each classifier is analyzed through accuracy and computational time. The entire framework is simulated on MATLAB2022a using a desktop computer with 16 GB of Ram and 8 GB graphics card.

### 3.1. CUI WCE Dataset Results

The classification results of proposed framework on CUI WCE dataset has been presented in Table 1. This table presented the results in four different experiments. In the first experiment (Org-MobV2), the fine-tuned MobileNet-V2 deep model is trained on original dataset and extract features that are later utilized for the classification. In the second experiment (Enh-MobV2), enhanced WCE images have been utilized and passed to fine-tuned model for training that were later utilized for features extraction and classification. In the third experiment (Seg-MobV2), localized infected images have been fed to fine-tune MobileNet-V2 for training that were later utilized for features extraction and classification. In the last experiment (Proposed), the entire proposed framework is utilized and classification was performed.

In Table 1, the maximum obtained accuracy of first experiment called Org-Mobv2 is 95.24% on ELM classifier whereas the minimum computational time is 116.5424 s. Moreover, the lowest accuracy of this experiment is 90.56% for fine-tree classifier. The best accuracy of the second experiment (Enh-MobV2) is 96.94% on ELM, whereas the lowest accuracy is 91.06% on fine-tree. The minimum computational time of this experiment is 110.2010 s, whereas the highest noted time is 141.5624 s. In the third experiment (Seg-MobV2), the maximum obtained accuracy is 97.39% that improved compared to the first two experiments. Moreover, it is also noted that computational time is little decreased after this experiment. For the proposed framework, the maximum obtained accuracy is 99.61% that is significantly improved compared to the first three experiments. Moreover, the computational time is significantly reduced after this step due to the use of optimization algorithm. In addition, Figure 8 illustrated a confusion matrix of ELM classifier that can be utilized for the verification of proposed accuracy. Hence, overall proposed framework obtained improved accuracy and consumes less time compared to the previous experiments. 

### 3.2. KVASIR V1 Dataset Results

The classification results of Kvasir V1 dataset has been presented in Table 2. Similar to results of CUI WCE dataset as presented in Section 3.1, the four experiments have been performed on Kvasir V1 dataset. In the first experiment, the maximum achieved accuracy is 95.24% whereas the minimum noted computational time is 92.1124 s. In the second experiment, the best obtained accuracy is 95.80% on ELM classifier. Compared to the first experiment, the current accuracy is little improved and also the computational time is reduced from 92.1124 s to 90.3645 s. The best obtained accuracy of third experiment is 96.14% on ELM classifier that is improved compared to the first two experiments. In addition, the computational time is reduced from 92.1123 s to 84.1046 s. The proposed framework obtained maximum accuracy of 98.20%, which is improved compared to the previous three experiments. Moreover, the lowest accuracy of this experiment is 93.02% W-KNN classifier. Moreover, the computational time of proposed framework is significantly reduced. Figure 9 shows the confusion matrix of ELM classifier for the proposed framework that can be utilized for the verification of the obtained accuracy of 98.20%. Based on the results, we can determine that the proposed framework performed better in both accuracy and computational time. 

### 3.3. KVASIR V2 Classification Results

The classification results of proposed framework on Kvasir V2 dataset have been presented in Table 3. Similar to Table 1 and Table 2, four experiments have been performed for this dataset to analyze the entire framework. For experiment 1, 94.20% accuracy is obtained on ELM classifier that is better compared to the other classifiers listed in this table. The lowest obtained accuracy of this experiment is 92.14% on fine-tree classifier. In addition, the computational time is also noted for each classifier and minimum time is 191.5246 s. In the second experiment, 95.18% best accuracy is obtained on ELM that is improved compared to the first experiment. This shows the importance of contrast enhancement step. In the third experiment, 96.76% accuracy is achieved that is better compared to the first two experiments. Moreover, this experiment consumes less time i.e., 173.1142 s, compared to the other listed experiments. Finally, proposed framework is employed and obtained an accuracy of 98.02% that is significantly improved compared to the other experiments. Moreover, the computational time of this experiment is far better than the other experiments on all classifiers. Figure 10 illustrates the confusion matrix of Kvasir V2 dataset on ELM classifier that can be utilized for the verification of the proposed framework accuracy. 

### 3.4. Discussion and Comparison

A brief discussion of proposed framework is also presented in terms of some visual facts and quantitative values. Figure 5 shows the qualitative results of infected lesion localization using proposed saliency based segmentation. From this figure, it is clearly illustrated that the proposed framework correctly segmented both larger and smaller infected regions. The quantitative results of all selected datasets have been presented under Table 1, Table 2 and Table 3 and confusion matrices in Figure 8, Figure 9 and Figure 10. From these figures, it is observed that the accuracy is improved by employing the proposed algorithm compared to the other mentioned methods such as Org-MobV2, Enh-MobV2, and Seg-MobV2. We also performed a *t*-test [34,35] and presented a hypothesis in which we assume that the accuracy will not degrade after employing the proposed algorithm (h = 1) and accuracy will degrade after employing the proposed algorithm (h = 0). We consider only ELM classifier values for all three selected datasets. For CUI WCE dataset, the performance results for ELM classifier were highly significant such as $p=1.74\times 10^{\left(-6\right)}$ for *t*-test. For Kvasir V1 dataset, the *p* value is computed by *t*-test and shows that the performance of ELM classifier is significant such as $p=6.63\times 10^{\left(-7\right)}$. Similarly, the *t*-test is peformed for Kvasir V2 dataset and obtained *p* value of $p=1.50\times 10^{\left(-6\right)}$ that shows the significance performance of ELM classifier for proposed framework. 

Moreover, a comparison is also conducted of proposed framework with some other deep neural nets such as VGG16, VGG19, AlexNet, and ResNet50. A few quantitative facts are presented in Table 4. Based on this table, it clearly proves the better performance of proposed framework. Finally, we compare the proposed framework accuracy with some recent techniques such as Khan et al. [36]. In this technique, the authors used a deep learning-based framework and achieved accuracies of 99.42, 97.85, and 97.2%, for CUI WCE dataset, Kvasir V1, and Kvasir V2, respectively. The proposed framework achieved accuracies of 99.61, 98.20, and 98.02%, respectively. 

## 4. Conclusions

This article proposes a new deep saliency estimation and Bayesian Optimization learning-based framework for detecting and classifying GIT diseases. The experiment was carried out on three publicly available datasets and yielded accuracies of 99.61, 98.20, and 98.02%, which were better compared to the previous method. The classification accuracy was improved by the contrast enhancement step. This step also increases the likelihood of correctly locating the infected region in the image. MobileNet-V2, a pre-trained deep model, is chosen and trained using Bayesian optimization and deep transfer learning. The benefit of this step was improved hyperparameter initialization, which was used for fine-tuning model training. In addition, we proposed a hybrid optimization algorithm for selecting the best features. This algorithm selects the best features to improve classification accuracy while decreasing computational time. The contrast enhancement and hyperparameter optimization were the work’s strengths. Furthermore, feature optimization reduced irrelevant information. As a future work, the contrast enhanced images will be passed to CNN models such as UNET and MASK RCNN for infected lesion segmentation. Moreover, weights of CNN models will be optimized through evolutionary optimization techniques.

## Figures and Tables

**Figure 1 diagnostics-12-02718-f001:**
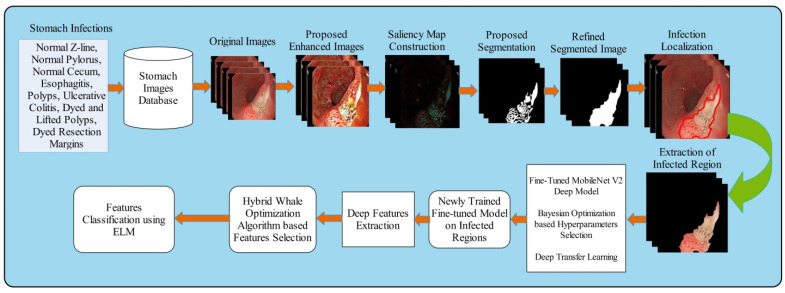
Proposed saliency estimation and deep learning framework for GIT diseases classification.

**Figure 2 diagnostics-12-02718-f002:**
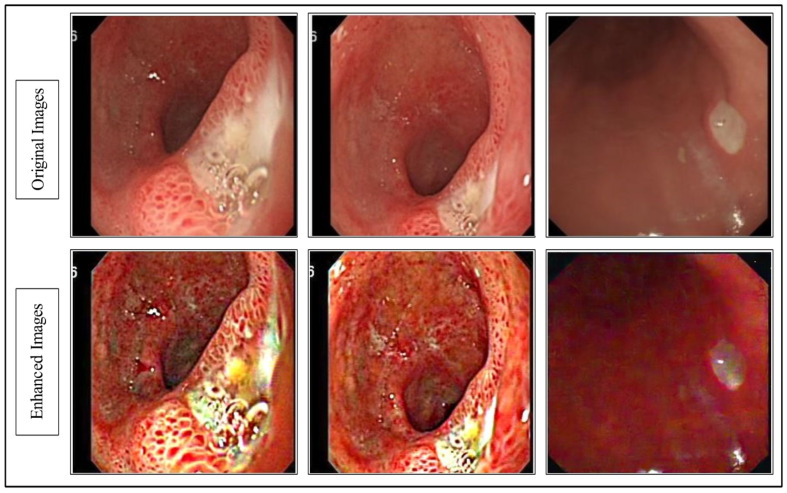
Proposed hybrid contrast enhancement image effects.

**Figure 3 diagnostics-12-02718-f003:**
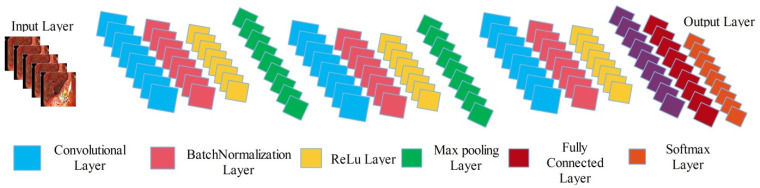
Proposed 14-layered CNN architecture for saliency map construction.

**Figure 4 diagnostics-12-02718-f004:**
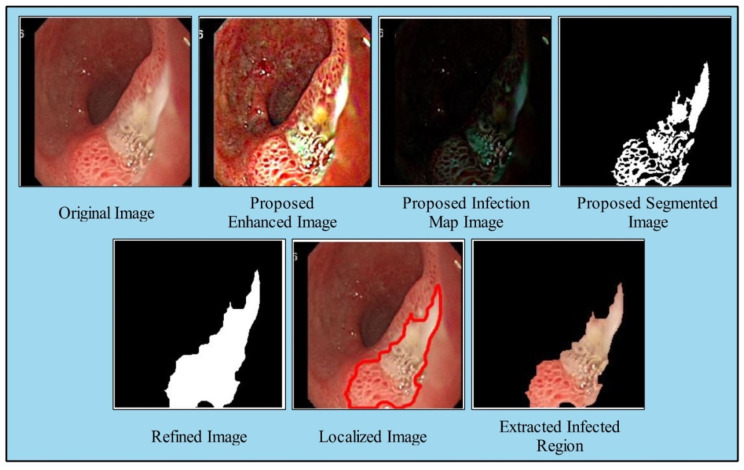
Proposed saliency based infected region detection.

**Figure 5 diagnostics-12-02718-f005:**
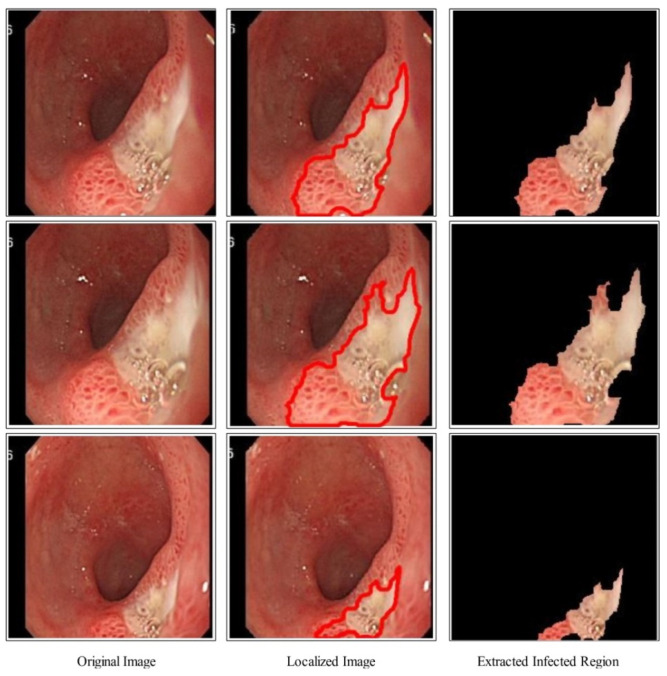
Proposed ulcer localization using deep saliency map.

**Figure 6 diagnostics-12-02718-f006:**
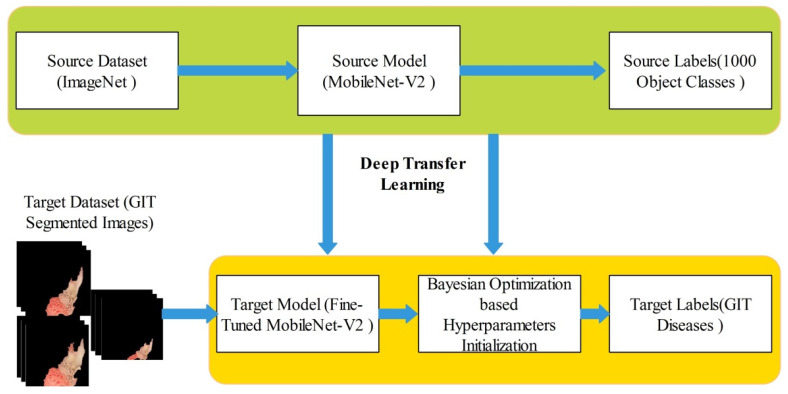
Deep TL and BO-based training of fine-tuned model on segmented GIT disease images.

**Figure 7 diagnostics-12-02718-f007:**
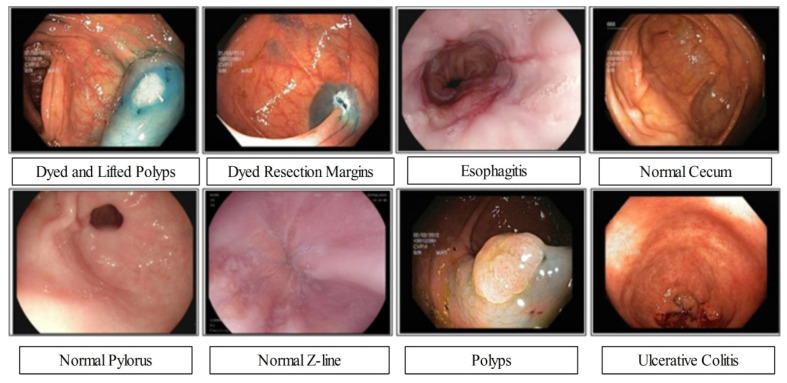
Samples GIT disease images collected from Kvasir-V2 dataset [32].

**Figure 8 diagnostics-12-02718-f008:**
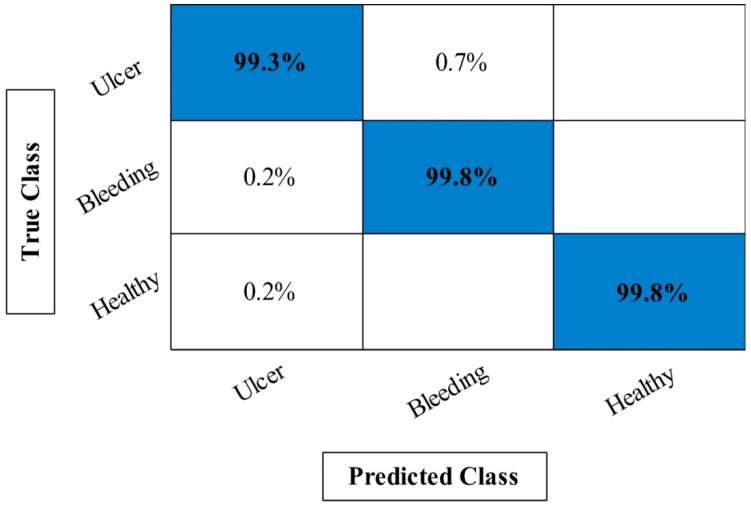
Confusion matrix of proposed framework for ELM classifier on CUI WCE dataset.

**Figure 9 diagnostics-12-02718-f009:**
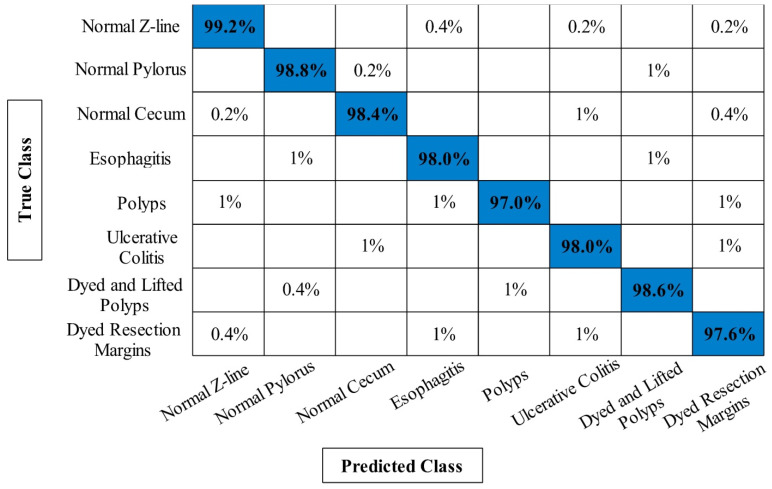
Confusion matrix of proposed framework for ELM classifier on Kvasir V1 dataset.

**Figure 10 diagnostics-12-02718-f010:**
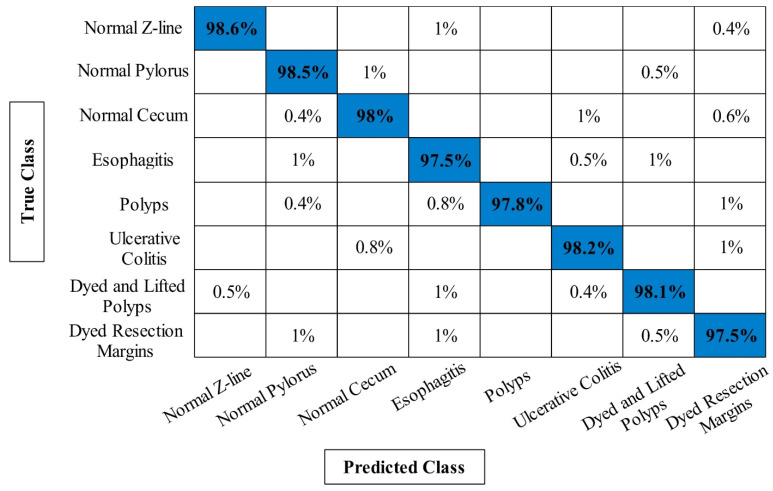
Confusion matrix of proposed framework for ELM classifier on Kvasir V2 dataset.

**Table 1 diagnostics-12-02718-t001:** Proposed framework classification results on CUI WCE dataset.

Classifier	Org-MobV2	Enh-MobV2	Seg-MobV2	Proposed	Accuracy (%)	Time (s)
	√				95.24	116.5424
**ELM**		√			96.94	110.2010
			√		97.39	103.1152
				√	**99.61**	**69.5442**
	√				90.56	131.5032
Fine Tree		√			91.04	122.5629
			√		92.39	116.0424
				√	94.84	74.5006
	√				92.10	147.0302
Q-SVM		√			94.56	141.5624
			√		94.90	136.9206
				√	96.36	89.1432
	√				91.04	134.1142
W-KNN		√			92.50	129.5260
			√		94.14	121.1124
				√	94.80	80.5142
	√				92.52	119.4504
Bi-Layer NN		√			94.06	113.1492
			√		95.84	106.5824
				√	98.16	71.0062
XGBOOST	√				92.26	126.0047
		√			92.95	110.8072
			√		93.60	106.4921
				√	94.00	70.6075

**Table 2 diagnostics-12-02718-t002:** Proposed framework classification results on Kvasir V1 dataset.

Classifier	Org-MobV2	Enh-MobV2	Seg-MobV2	Proposed	Accuracy (%)	Time (s)
	√				95.24	92.1124
**ELM**		√			95.80	90.3645
			√		96.14	84.1046
				√	**98.20**	**52.5046**
	√				93.04	98.2446
**Fine Tree**		√			93.64	95.3604
			√		94.03	87.2942
				√	96.46	63.1142
	√				91.62	97.3046
**Q-SVM**		√			92.14	94.2946
			√		93.42	87.5042
				√	95.14	71.0246
	√				90.54	99.6214
**W-KNN**		√			91.32	97.5429
			√		91.98	90.1120
				√	93.02	61.1129
	√				94.34	94.1126
**Bi-Layer NN**		√			94.96	91.6624
			√		95.70	86.2404
				√	97.10	55.5246
XGBOOST	√				93.58	104.5093
		√			94.02	101.9226
			√		95.16	93.5521
				√	95.90	69.4050

**Table 3 diagnostics-12-02718-t003:** Proposed framework classification results on Kvasir V2 dataset.

Classifier	Org-MobV2	Enh-MobV2	Seg-MobV2	Proposed	Accuracy (%)	Time (s)
	√				94.20	191.5246
**ELM**		√			95.18	186.5509
			√		96.76	173.1142
				√	**98.02**	**102.5026**
	√				92.14	205.0426
Fine Tree		√			93.24	201.0020
			√		93.60	191.5462
				√	95.46	120.2500
	√				92.92	226.2042
Q-SVM		√			93.60	216.1120
			√		94.10	204.0526
				√	96.40	140.0329
	√				92.62	207.1246
W-KNN		√			92.94	203.0204
			√		93.56	195.5509
				√	95.84	120.5426
	√				93.04	195.0694
Bi-Layer NN		√			94.16	191.1124
			√		94.86	185.0329
				√	96.94	110.0046
XGBOOST	√				93.00	220.0945
		√			93.84	211.2572
			√		94.10	182.9443
				√	94.90	136.0790

**Table 4 diagnostics-12-02718-t004:** Comparison of proposed framework with other neural nets using selected datasets.

Method	Accuracy (%)		
	CUI WCE Dataset	Kvasir V1 Dataset	Kvasir V2 Dataset
VGG16 Model embedded in Figure 1 instead of MobileNetV2	95.80	94.28	95.11
VGG19 Model embedded in Figure 1 instead of MobileNetV2	96.24	95.02	95.76
AlexNet Model embedded in Figure 1 instead of MobileNetV2	95.16	94.66	94.90
ResNet50 Model embedded in Figure 1 instead of MobileNetV2	97.88	95.89	96.16
Khan et al. [36], 2022	99.42	97.85	97.85
**Proposed Framework**	**99.61**	**98.20**	**98.02**

## Data Availability

The datasets used in this work are publicly available.

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
