# Peer review of "GestroNet: A Framework of Saliency Estimation and Optimal Deep Learning Features Based Gastrointestinal Diseases Detection and Classification"

_diagnostics, 2022, doi:10.3390/diagnostics12112718_

Round 1
Reviewer 1 Report
The paper presents a learning-based approach for the classification of gastrointestinal diseases. Compared to other approaches, the presented approach improves the classification performance. The paper is well-written and experimental results demonstrate the effectiveness of the approach. However, the following comments need to be addressed:
-Deep transfer learning in Figure 6 needs to be detailed.
For the pre-trained MobileNet, you need to specify if some layers
were (un)Frozen?. Also, the MobileNet-V2 was pre-trained on Imagenet (source data) that is not closely related to the gastrointestinal image datasets (target data). Need to train MobileNet-V2 on other related source data and to report performance
-In Figures 8-10, need to show numbers and percents. Also, results reported on the whole target data?
- Utilizing XGBOOST classifier
-Report results in Tables 1-4 using other performance measures in multiclass classification such as precision, Recall, and F1 (Macro- (and Micro-) averaging).
-Report p-values and assess if the proposed method yields significant results.
Author Response
Response sheet attached. thank you

Reviewer 2 Report
This paper is a great effort of the authors. However, I appreciate a few more improvements:
1. 1. The problem statement needed to be more precise in the abstract.
2. 2. The contribution part in the Introduction shall be well-identified and presented in a specific manner.
3. 3. In the introduction section, a few more pieces of literature can be great for the authors. Recent reference is required to improve the Introduction part. As an example, for deep learning structure for EEG i.e., DOI: 10.3390/app10217639 can be a suitable resource to review.
4. 4. The dataset section should be improved. The details of the dataset are expected to be discussed.
5. 5. Future directions should be well highlighted in the conclusion.
Author Response
Response sheet attached. thank you

Round 2
Reviewer 1 Report
Authors addressed points except 1 point (i.e., "C6: Report p-values and assess if the proposed method yields significant results") that needs to be addressed properly.
Kindly see the following references to address the comment:
Demšar, Janez. "Statistical comparisons of classifiers over multiple data sets." The Journal of Machine learning research 7 (2006): 1-30. Calvo, Borja, and Guzmán Santafé Rodrigo. "scmamp: Statistical comparison of multiple algorithms in multiple problems." The R Journal, Vol. 8/1, Aug. 2016 (2016). For example, using Friedman test with the Bergmann-Hommel post hoc procedure
Author Response
Response sheet added. thanks

Reviewer 2 Report
Congratulations.
Author Response
Thank you for your valuable recommendation. We once again carefully revised the entire manuscript in terms of grammar and typos as a final version.
Round 3
Reviewer 1 Report
In terms of updates in Lines 486-504, you need to show ONLY the p-values obtained from the statistical test. Kindly Remove other values. And use scientific notations for P-values
For example,
For CUI WCE dataset,
the performance results for ELM classifier were highly significant
(P = 1.74 x 10^-6 from a t-test)
Do the same for others
Author Response
Response sheet attached, thanks
